# A 2-PARAMETER PERSISTENCE LAYER FOR LEARNING

## ABSTRACT

1-parameter persistent homology, a cornerstone in Topological Data Analysis (TDA), studies the evolution of topological features such as connected components and cycles hidden in data. It has found its application in strengthening the representation power of deep learning models like Graph Neural Networks (GNN). To enrich the representations of topological features, here we propose to study 2-parameter persistence modules induced by bi-filtration functions. In order to incorporate these representations into machine learning models, we introduce a novel vectorization on 2-parameter persistence modules called Generalized Rank Invariant Landscape (GRIL). We show that this vector representation is 1-lipschitz (stable) and differentiable with respect to underlying filtration functions and can be easily integrated into machine learning models to augment encoding topological features. We present an algorithm to compute the vectorization and its gradients. We also test our methods on synthetic graph datasets and benchmark graph datasets, and compare the results with previous vector representations of 1-parameter and 2-parameter persistence modules

## 1 INTRODUCTION

Machine learning models such as and Graph Neural Networks (GNNs) (Gori et al., 2005; Scarselli et al., 2009; Kipf & Welling, 2017; Xu et al., 2019) are well-known successful tools from the geometric deep learning community. The representation power of such models can be augmented by infusing topological information as some vector representation of persistent homology of the underlying space hidden in data. Many recent works have successfully integrated topological information with machine learning models. (Carrière et al., 2020; Kim et al., 2020; Gabrielsson et al., 2020; Hofer et al., 2020; Horn et al., 2022; Swenson et al., 2020; Bouritsas et al., 2022; Corbet et al., 2019; Carrière & Blumberg, 2020; Vipond, 2020). In most of these works, the authors use 1-parameter persistence homology as the topological information. However, in (Corbet et al., 2019; Vipond, 2020; Carrière & Blumberg, 2020), the authors use vector representations of 2-parameter persistence modules. In (Carrière & Blumberg, 2020) and (Corbet et al., 2019), these representations are based on slices of 2-parameter persistence modules along lines, which are first studied and computed by (Lesnick & Wright, 2015). In (Vipond, 2020), the author generalizes the notion of 1-parameter persistence landscapes (Bubenik, 2015). In this paper we propose a novel vector representation *Generalized Rank Invariant Landscape* (GRIL) for 2-parameter persistence modules which encodes richer information beyond fibered barcodes alone. The building blocks are based on the idea of *generalized rank invariant* (Kim & Mémoli, 2021; Dey et al., 2022). The construction of GRIL is a generalization of persistence landscape (Bubenik, 2015; Vipond, 2020). We will show that the vector representation GRIL is 1-Lipschitz and differentiable with respect to the filtration function $f$, which allows us to build a differentiable topological layer, PERSGRIL, in a machine learning pipeline. We demonstrate its use on synthetic datasets and standard graph datasets [1]. From the perspective of direct use of 2-parameter persistence modules into machine learning models, to the best of our knowledge, this is the first work of its kind.

Persistent homology is a useful tool for characterizing the shape of data. Rooted in the theory of algebraic topology and algorithms, it has spawned the flourishing area of Topological Data Analysis(TDA). The classical persistent homology, also known as, 1-parameter persistence module, has attracted plenty of attention from both theory (Edelsbrunner & Harer, 2010; Oudot, 2015; Carlsson & Vejdemo-Johansson, 2021; Dey & Wang, 2022; Hofer et al., 2017; Li et al., 2022; Dey & Wang,

---

[1] the code for full implementation will be available after review process is completed.

2022; Mémoli et al., 2022) and applications (Yan et al., 2021; Zhao et al., 2020; Yang et al., 2021b;a; Banerjee et al., 2020; Wu et al., 2020; Wang et al., 2020; Chen et al., 2021; Hu et al., 2021; Yan et al., 2022). The standard pipeline of 1-parameter persistence module is as follows: Given a domain of interest $\mathcal{X}$ (e.g. a topological space, point cloud data, a graph, or a simplicial complex) with a scalar function $f : \mathcal{X} \to \mathbb{R}$, one filters the domain $\mathcal{X}$ by the sublevel sets $\mathcal{X}_\alpha \triangleq \{x \in \mathcal{X} \mid f(x) \leq \alpha\}$ along with a continuously increasing threshold $\alpha \in \mathbb{R}$. The collection $\{\mathcal{X}_\alpha\}$, which is called a *filtration*, forms an increasing sequence of subspaces $\emptyset = \mathcal{X}_{-\infty} \subseteq \mathcal{X}_{\alpha_1} \subseteq \cdots \subseteq \mathcal{X}_{+\infty} = \mathcal{X}$. Along with the filtration, topological features appear, persist, and disappear over some intervals. We consider $p$-homology groups $H_p(-)$ (over a field, see (Hatcher, 2000)) of the subspaces in this filtration, which results into a sequence of vector spaces. These vector spaces are connected by inclusion-induced linear maps forming an algebraic structure $0 = H_p(\mathcal{X}_{-\infty}) \to H_p(\mathcal{X}_{\alpha_1}) \to \cdots \to H_p(\mathcal{X}_{+\infty})$. (Hatcher, 2000)). This algebraic structure, known as 1-parameter persistence module induced by $f$ and denoted as $M^f$, can be uniquely decomposed into a collection of atomic modules called interval modules, which completely characterizes the topological features in regard to the three behaviors–appearance, persistence, and disappearance of all $p$-dimensional cycles. This unique decomposition of 1-parameter persistence module is commonly summarized as a *persistence diagram* (Edelsbrunner et al., 2002) or *barcode* (Zomorodian & Carlsson, 2005). Figure 1 (left) shows a filtration of a simplicial complex which induces a 1-parameter persistence module and its decomposition into bars.

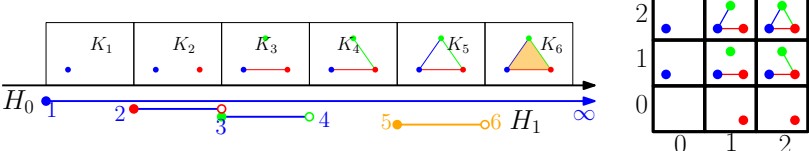

Figure 1: (left) 1-parameter filtration and bars; (right) a 2-parameter filtration inducing a 2-parameter persistence module whose decomposition is not shown.

Some problems in practice may demand tracking the topological information in a filtration that is not necessarily linear. For example, in (Adcock et al., 2014), 2-parameter persistence module is shown to be better for classifying hepatic lesions compared to 1-parameter persistence. In (Keller et al., 2018), a virtual screening system based on 2-parameter persistence modules are shown to be effective for searching new candidate drugs. In such applications, instead of studying a sequential filtration filtered by a scalar function, one may study a grid-filtration induced by a $\mathbb{R}^2$-valued bi-filtration function $f : \mathcal{X} \to \mathbb{R}^2$ with $\mathbb{R}^2$ equipped with partial order $\boldsymbol{u} \leq \boldsymbol{v} : u_1 \leq v_1, u_2 \leq v_2$; see Figure 1(right) for an example of 2-parameter filtration. Following a similar pipeline as the 1-parameter persistence module, one will get a collection of vector spaces $\{M^f_{\boldsymbol{u}}\}_{\boldsymbol{u} \in \mathbb{R}^2}$ indexed by vectors $\boldsymbol{u} = (u_1, u_2) \in \mathbb{R}^2$ and linear maps $\{M^f(\boldsymbol{u} \leq \boldsymbol{v}) : M^f_{\boldsymbol{u}} \to M^f_{\boldsymbol{v}} \mid \boldsymbol{u} \leq \boldsymbol{v} \in \mathbb{R}^2\}$ for all comparable $\boldsymbol{u} \leq \boldsymbol{v}$. The entire structure $M^f$, in analogy to the 1-parameter case, is called a 2-parameter persistence module induced from $f$. Unlike 1-parameter case, the algebraic structure of 2-parameter persistence modules is much more complicated. There is no *complete* discrete invariant like persistence diagrams or barcodes for 2-parameter persistence modules (Carlsson & Zomorodian, 2009). A good non-complete invariant for 2-parameter persistence modules should characterize as many non-isomorphic topological features as possible. At the same time it should be stable with respect to small perturbations of filtration functions, which guarantees its important properties of continuity and differentiability for machine learning models. Therefore, how to build a good summary for 2-parameter persistence modules which is also applicable to machine learning models is an important problem.

## 2 2-PARAMETER PERSISTENCE LANDSCAPE

From the perspective of representation learning, a persistence module can be viewed as a special representation of a discrete topological space, like point cloud data or graph embedding, which captures geometric and topological information. 1-parameter persistence module captures information about topological features that persist across different scales. Here, we consider a bi-filtration which leads to a 2-parameter persistence module. To better utilize the richer information captured by 2-

parameter persistence modules, here we propose GRIL (Generalized Rank Invariant Landscape), a stable and differentiable vectorized representation of a 2-parameter persistence module.

Let $M = M^f$ be a 2-parameter persistence module induced by a filtration function $f$. We say a connected subset $I \subseteq \mathbb{R}^2$ is an *interval* if $\forall \boldsymbol{u} \le \boldsymbol{v} \le \boldsymbol{w}, [\boldsymbol{u} \in I, \boldsymbol{w} \in I] \implies [\boldsymbol{v} \in I]$. The restriction of $M$ to an interval $I$, denoted as $M|_I$, is the collection of vector spaces $\{M_{\boldsymbol{u}} \mid \boldsymbol{u} \in I\}$ along with linear maps $\{M(\boldsymbol{u} \le \boldsymbol{v}) \mid \boldsymbol{u}, \boldsymbol{v} \in I\}$. One can define generalized rank:

$$\mathsf{rk}^M(I) \triangleq \mathrm{rank}[\varprojlim M|_I \to \varinjlim M|_I]$$

where $\varprojlim M|_I \to \varinjlim M|_I$ is the unique linear map from the limit of $M|_I$ to the colimit of $M|_I$. When $I = [\boldsymbol{u}, \boldsymbol{v}] \triangleq \{\boldsymbol{w} \in \mathbb{R}^2 \mid \boldsymbol{u} \le \boldsymbol{w} \le \boldsymbol{v}\}$ is a rectangle subset in $\mathbb{R}^2$, $\varprojlim M|_I = M_{\boldsymbol{u}}$ and $\varinjlim M|_I = M_{\boldsymbol{v}}$. Then $\mathsf{rk}^M(I)$ equals the traditional rank of the linear map $M(\boldsymbol{u} \le \boldsymbol{v})$. We refer the reader to (MacLane, 1971) for the definitions of limit and colimit in category theory. The basic idea of GRIL is to compute a collection of generalized ranks $\{\mathsf{rk}^M(I)\}_{I \in \mathcal{W}}$ over some covering set $\mathcal{W}$ on $\mathbb{R}^2$, which is called a *generalized rank invariant* (Kim & Mémoli, 2021) of $M$ over $\mathcal{W}$.

We choose $\mathcal{W}$ to be a set of *Worms* defined as follows:

$$\mathcal{W} \triangleq \left\{ \boxed{\boldsymbol{p}}_\delta^\ell \mid \delta > 0, \ell \ge 1, \boldsymbol{p} \in \mathbb{R}^2 \right\}$$

where

$$\boxed{\boldsymbol{p}}_\delta^\ell \triangleq \{\boldsymbol{q} \mid \exists \alpha \in \mathbb{R}, |\alpha| \le (\ell-1)\delta : \|\boldsymbol{q} - \boldsymbol{p} - (\alpha, -\alpha)\|_\infty \le \delta\}.$$

We call the $\boldsymbol{p}$ in $\boxed{\boldsymbol{p}}_\delta^\ell$ the *center point* of the $\ell$-worm and $\delta$ the *width* of the $\ell$-worm. As a special case, when $\ell = 1$, $\boxed{\boldsymbol{p}}_\delta^1 = \boxed{\boldsymbol{p}}_\delta \triangleq \{\boldsymbol{q} : \|\boldsymbol{p} - \boldsymbol{q}\|_\infty \le \delta\}$ is a $\delta$-square with side $2\delta$ centered at $\boldsymbol{p}$. In general, for any $\ell \ge 1$, $\delta > 0$, $\ell$-worm $\boxed{\boldsymbol{p}}_\delta^\ell$ is the union of all $\delta$-squares $\boxed{\boldsymbol{q}}_\delta$ centered at some point $\boldsymbol{q}$ on the off-diagonal line segment $\boldsymbol{p} + \alpha(1, -1)$ with $|\alpha| \le (\ell-1)\delta$. Therefore, we can also equivalently write $\boxed{\boldsymbol{p}}_\delta^\ell = \bigcup_{\substack{\boldsymbol{q} = \boldsymbol{p} + (\alpha, -\alpha) \\ |\alpha| \le (l-1)\delta}} \boxed{\boldsymbol{q}}_\delta$. See Figure 2 (left) for an illustration of a 2-worm example. We now define GRIL.

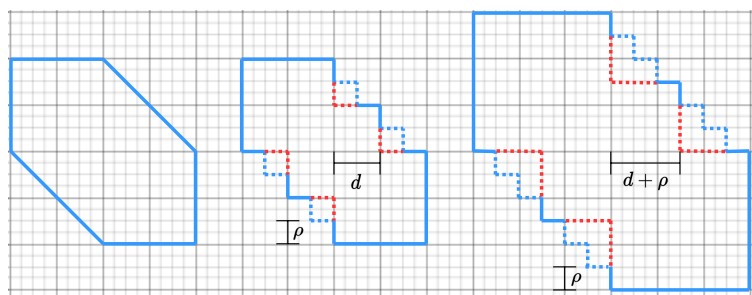

Figure 2: A 2-worm, discretized 2-worm and expanded discretized 2-worm. $\rho$ denotes grid resolution. The blue dotted lines show the intermediate staircase with step-size $\rho$. The red dotted lines form parts of the squares with size $d$ which are replaced by the blue dotted lines in the worm. The last figure shows the expanded 2-worm with red and blue dotted lines. The expanded 2-worm has width $d + \rho$ which is the one step expansion of the worm with width $d$.

**Definition 2.1** (Generalized Rank Invariant Landscape ). For a persistence module $M$, the *Generalized Rank Invariant Landscape (GRIL)* of $M$ is a function $\lambda^M : \mathbb{R}^2 \times \mathbb{N}_+ \times \mathbb{N}_+ \to \mathbb{R}$ defined as

$$\lambda^M(\boldsymbol{p}, k, \ell) \triangleq \sup_{\delta \ge 0} \{\mathsf{rk}^M(\boxed{\boldsymbol{p}}_\delta^\ell) \ge k\}. \tag{1}$$

**Proposition 2.1.** GRIL *is equivalent to the generalized rank invariant on $\mathcal{W}$. Here the equivalence means bijective reconstruction from each other (proof in Appendix B).*

In practice, we choose center points $\boldsymbol{p}$ from some finite subset $\mathcal{P} \subset \mathbb{R}^2$, e.g. a finite uniform grid in $\mathbb{R}^2$, and consider $k \le K, \ell \le L$ for some $K, L \in \mathbb{N}_+$. Then $\lambda^M$ can be viewed as a vector of dimension $|\mathcal{P}| \times K \times L$. See Figure 3 for an illustration of the overall pipeline of our construction

of $\lambda^M$ starting from a filtration function on a simplicial complex. Figure 4 shows the discriminating power of GRIL where we see that GRIL can differentiate between shapes that are topologically non-isomorphic.

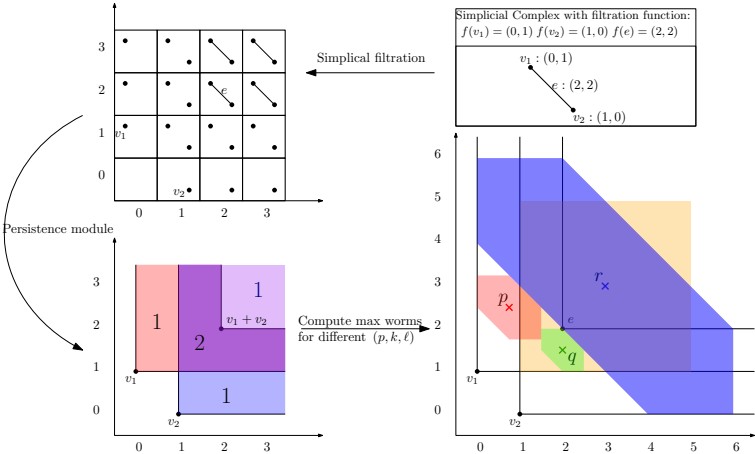

Figure 3: The construction starts from a simplicial complex with a bi-filtration function as shown on the top right. The simplicial complex consists of two vertices connected by one edge. Based on the bi-filtration, a simplicial bi-filtration can be defined as shown on the top left. On the bottom left, a 2-parameter persistence module is induced from the above simplicial filtration. If we check the dimensions of the vector spaces on all points of the plane, there are 1-dimensional vector spaces on red, blue and light purple regions. On the $L$-shaped dark purple region, the vector spaces have dimension 2. Finally, on this 2-parameter persistence module, we calculate $\lambda^{M^f}(\boldsymbol{p}, k, \ell)$ for all tuples $(\boldsymbol{p}, k, \ell) \in \mathcal{P} \times K \times L$ to get our GRIL vector representation. By Defintion 2.1 the value $\lambda^{M^f}(\boldsymbol{p}, k, \ell)$ corresponds to the width of the supremum $\ell$-worm on which the generalized rank is at least $k$. On the bottom right, the interval in red is the maximal 2-worm for $\lambda^{M^f}(\boldsymbol{p}, k = 1, \ell = 2)$. The green interval is the maximal 2-worm for $\lambda^{M^f}(\boldsymbol{q}, k = 2, \ell = 2)$. The yellow square is the maximal 1-worm for $\lambda^{M^f}(\boldsymbol{r}, k = 1, \ell = 1)$, and the blue interval is the maximal 3-worm for $\lambda^{M^f}(\boldsymbol{r}, k = 1, \ell = 3)$.

**Stability and Differentiability of GRIL.** An important property of GRIL is its stability property which makes it immune to small perturbations of the input bi-filtration while still retaining the ability to characterize topologies. We show GRIL is 1-Lipschitz (stable) with respect to input filtrations.

**Proposition 2.2.** *Given two filtration functions* $f, f' : \mathcal{X} \to \mathbb{R}^2$, $\|\lambda^{M^f} - \lambda^{M^{f'}}\|_\infty \leq \|f - f'\|_\infty$ *(proof in Appendix B).*

**Remark 2.1.** Note that when $\mathcal{X}$ is a finite space (e.g. finite simplicial complex (see Definition A.1), point cloud) with $|\mathcal{X}| = n$ then, any $f \colon \mathcal{X} \to \mathbb{R}^2$ can be represented as a vector in $\mathbb{R}^{2n}$.

We now define PERSGRIL.

**Definition 2.2** (PERSGRIL). For a finite space $\mathcal{X}$ with $|\mathcal{X}| = n$ and fixed $k, \ell, \boldsymbol{p}$, PERSGRIL is a function $\Lambda_{\boldsymbol{p}}^{k,\ell} : \mathbb{R}^{2n} \to \mathbb{R}$ given by $\Lambda_{\boldsymbol{p}}^{k,\ell}(f) = \lambda^{M^f}(k, \ell, \boldsymbol{p})$.

**Proposition 2.3.** *PERSGRIL is Lipschitz continuous with respect to the bi-filtration functions on finite spaces. (proof in Appendix B.)*

---

[2]$y$-axis represents Rips filtration and $x$-axis represents density. Density value of a vertex $v$, denoted by $\gamma_v$, is defined as $1 - \exp($Avg. nearest neighbor distance of $v)$. For an edge $e := (u, v)$ and a triangle $t := (u, v, w)$, $\gamma$ is defined as $\max(\gamma_u, \gamma_v)$ and $\max(\gamma_u, \gamma_v, \gamma_w)$ respectively. Rips filtration value of all vertices $r_v$ is 0. For an edge $e := (u, v)$, the Rips filtration value $r_{uv}$ is the Euclidean distance between $u$ and $v$. For a triangle $t := (u, v, w)$, the filtration value $r_t = \max(r_{uv}, r_{uw}, r_{vw})$ is the maximum over all its edges. The filtration value is rounded off to the nearest hundredth decimal place for visualization purposes.

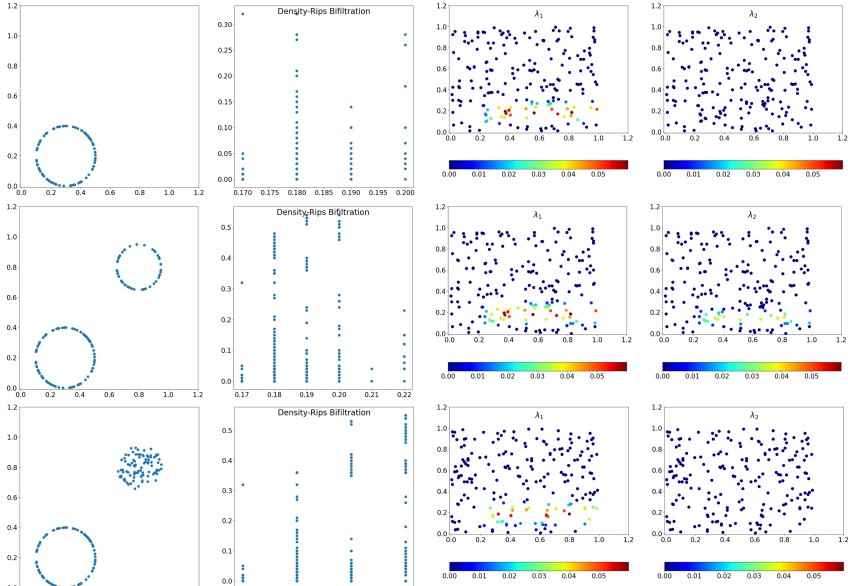

Figure 4: GRIL as a topological discriminator: each row shows a point cloud $P$, its density-Rips bi-filtration [2], GRIL value heatmap for 1-dimensional homology and generalized ranks $k = 1$ and $k = 2$ named as $\lambda_1$ and $\lambda_2$ respectively. First Betti number ($\beta_1$) of a circle is 1 which is reflected in $\lambda_1$ being non-zero. $\beta_1$ for two circles is 2 which is reflected in both $\lambda_1$ and $\lambda_2$ being non-zero. Similarly, $\beta_1$ of a circle and disk together is 1 which is reflected in $\lambda_1$ being non-zero but $\lambda_2$ being zero for this point cloud.

By Rademacher's theorem (Evans & Gariepy, 2015), we have PERSGRIL, as a Lipschitz continuous function, being differentiable almost everywhere.

**Corollary 2.4.** PERSGRIL is differentiable almost everywhere.

The differentiability of PERSGRIL in Corollary 2.4 refers to the existence of all directional derivatives. But the existence of a steepest direction as the "gradient" of PERSGRIL might not be unique. We propose an algorithm to efficiently compute one specific steepest direction based on the following theorem.

**Theorem 2.5.** *Consider the space of all filtration functions* $\{f : \mathcal{X} \to \mathbb{R}^2\}$ *on a finite space* $\mathcal{X}$ *with* $|\mathcal{X}| = n$, *which is equivalent to* $\mathbb{R}^{2n}$. *For fixed* $k, \ell, \boldsymbol{p}$, *there exists a measure-zero subset* $Z \subseteq \mathbb{R}^{2n}$ *such that for any* $f \in \mathbb{R}^{2n} \setminus Z$ *satisfying the following generic condition:* $\forall x \neq y \in \mathcal{X}, f(x)_1 \neq f(y)_1, f(x)_2 \neq f(y)_2$, *there exists an assignment* $s : \mathcal{X} \to \{\pm 1, 0, \pm \ell\}^2$ *such that*

$$\nabla_s \Lambda_{\boldsymbol{p}}^{k,\ell}(f) \triangleq \lim_{\alpha \to 0} \frac{\Lambda_{\boldsymbol{p}}^{k,\ell}(f + \alpha s) - \Lambda_{\boldsymbol{p}}^{k,\ell}(f)}{\alpha \|s\|_\infty} = \max_{g \in \mathcal{X}} \nabla_g \Lambda_{\boldsymbol{p}}^{k,\ell}(f).$$

The proof of Theorem 2.5 in Appendix B also shows how to find the assignment $s$ with the corresponding set of *supporting* simplices. This result leads us to update the simplicial filtration with such an assignment $s$. See the description of enhancing topological features in section 4.

## 3   ALGORITHM

We present an algorithm to compute GRIL in this section. High-level idea of the algorithm is as follows: Given a bi-filtration function $f : \mathcal{X} \to \mathbb{R}^2$, for each $(\boldsymbol{p}, k, \ell) \in \mathcal{P} \times K \times L$, we need to compute $\lambda^{M^f}(\boldsymbol{p}, k, \ell) = \sup_{\delta \geq 0}\{\mathsf{rk}^{M^f}(\boxed{\boldsymbol{p}}_\delta^\ell) \geq k\}$. In essence, we need compute the maximum width over worms on which the generalized rank is at least $k$. In order to find the value of this width, we use binary search. We compute generalized rank $\mathsf{rk}^{M^f}\left(\boxed{\boldsymbol{p}}_\delta^\ell\right)$ by applying the algorithm proposed in (Dey et al., 2022), which uses zigzag persistence on a boundary path. This zigzag

persistence is computed efficiently by a recent algorithm proposed in (Dey & Hou, 2022) [3]. We denote the sub-routine to compute generalized rank over a worm by COMPUTERANK in algorithm 1 mentioned below. COMPUTERANK($f, I$) takes as input a bi-filtration function $f$ and an interval $I$, and outputs generalized rank over that interval. In order to use the algorithm proposed in (Dey et al., 2022), the worms need to have their boundaries aligned with a grid structure defined on the range of $f$. Thus, we normalize $f$ to be in the range $[0, 1] \times [0, 1]$, define a grid structure on $[0, 1] \times [0, 1]$ and discretize the worms. Let $\text{Grid} = \{\left(\frac{m}{M}, \frac{n}{M}\right) \mid m, n \in \{0, 1, \dots, M\}\}$ for some $M \in \mathbb{Z}_+$. We denote the grid resolution as $\rho \triangleq 1/M$. We uniformly sample center points for the worms $\mathcal{P} \subseteq \text{Grid}$ from this grid. We consider discrete worms $\left[\hat{\boldsymbol{p}}\right]_\delta^\ell \triangleq \bigcup_{\substack{\boldsymbol{q}=\boldsymbol{p}+(\alpha,-\alpha) \\ |\alpha| \leq (l-1)\delta \\ \boldsymbol{q} \in \text{Grid}}} \left[\boldsymbol{q}\right]_\delta$ for all $\boldsymbol{p} \in \mathcal{P}$.

See Figure 2 as an illustration of discrete worms. Now all the discrete worms $\left[\hat{\boldsymbol{p}}\right]$ are intervals whose boundaries are aligned with the Grid. We apply the procedure COMPUTERANK$\left(f, \left[\hat{\boldsymbol{p}}\right]_\delta^\ell\right)$ to compute $\text{rk}^{M^f}\left(\left[\hat{\boldsymbol{p}}\right]_\delta^\ell\right)$. Let $\hat{\lambda}^{M^f}(\boldsymbol{p}, k, \ell) = \sup_{\delta \geq 0}\{\text{rk}^{M^f}(\left[\hat{\boldsymbol{p}}\right]_\delta^\ell) \geq k\}$. One can observe that $\hat{\lambda}^{M^f}(\boldsymbol{p}, k, \ell) - \lambda^{M^f}(\boldsymbol{p}, k, \ell) \leq \rho$. Therefore, we compute $\hat{\lambda}$ as an approximation of $\lambda$ with the approximation gap controlled by the grid resolution $\rho$. The pseudo-code is given in Algorithm 1. The algorithm is described in detail in Appendix C

---

**Algorithm 1** COMPUTEGRIL

---

**Input:** $f$ : Bi-filtration function, $\ell \geq 0, k \geq 1, \boldsymbol{p} \in \mathcal{P} \subseteq \text{Grid}$, $\rho$: grid resolution
**Output:** $\lambda(\boldsymbol{p}, k, l)$ : Persistence landscape at a point $\boldsymbol{p}$ for fixed $k$ and $\ell$
1: $d_{min} \leftarrow \rho, d_{max} \leftarrow 1$
2: **while** $d_{min} \leq d_{max}$ **do**
3:     $d \leftarrow (d_{min} + d_{max})/2$
4:     $I \leftarrow \left[\hat{\boldsymbol{p}}\right]_d^\ell$
5:     $r \leftarrow$ COMPUTERANK$(f, I)$
6:     **IF** $r = k$ **THEN**
7:         $\text{rk} \leftarrow d$
8:         $d_{min} \leftarrow d + \rho$
9:     **ELSE IF** $r > k$ **THEN**
10:         $d_{min} \leftarrow d + \rho$
11:     **ELSE IF** $r < k$ **THEN**
12:         $d_{max} \leftarrow d - \rho$
    **RETURN** $\text{rk}$

---

**Time complexity.** Assuming a grid with $t$ nodes and a bi-filtration of a complex with $n$ simplices on it, one can observe that each probe in the binary search takes $O((t + n)^\omega)$ time where $\omega < 2.37286$ is the matrix multiplication exponent (Alman & Williams, 2021). This is because each probe generates a zigzag filtration of length $O(t)$ with $O(n)$ simplices. Therefore, the binary search takes $O((t+n)^\omega \log t)$ time giving a total time complexity of $O(t((t+n)^\omega \log t))$ that accounts for $O(t)$ worms.

## 4 EXPERIMENTS

We create a differentiable topological layer based on GRIL named PERSGRIL which is in line with Definition 2.2. In essence, PERSGRIL takes in a bi-filtration function as input and gives the value of GRIL on the persistence module generated by the filtration function as output.

**Experiment with HourGlass dataset.** We test our model on a synthetic dataset (HourGlass) that entails a binary graph classification problem over a collection of attributed undirected graphs. Note that this synthetic dataset is designed to show that some attributed graphs can be easily classfied by 2-parameter persistence modules while not easy for 1-parameter persistence moduels or commonly

---
[3]https://github.com/taohou01/fzz

used GNN models. Each graph $G$ from either class is composed with two circulant subgraphs $G_1, G_2$ connected by some cross edges. The node attributes are order indices generated by two different traversals $T_1, T_2$. The label of classes corresponds to these two different traversals $T_1, T_2$. Therefore, the classification task is that given an attributed graph $G$, the model needs to predict which traversal is used to generate $G$. See Figure 5 (left) as an example of two attributed graphs with same graph structure but node attributes generated by two different traversals. More details can be found in Appendix D.1. We denote HourGlass[a,b] as the dataset of graphs generated with node size of each circulant subgraphs in range $[a, b]$. We generate three datasets with different sizes: HourGlass[10,20], HourGlass[21,30], HourGlass[31,40]. Each dataset contains roughly 400 graphs. We evenly split HourGlass[21,30] into balanced training set and testing set on which we compare PERSGRIL with several commonly used GNN models from the literature including: Graph Convolutional Networks (GCN)(Kipf & Welling, 2017), Graph Isomorphism Networks (GIN) (Xu et al., 2019) and a 1-parameter persistent homology vector representation called persistence image (PersImg (Adams et al., 2017). All GNN models contain 3 aggregation layers. All models use 3-layer multilayer perceptron (MLP) as classifiers. More details about model and training settings can be found in Appendix D.1. After that we also test these trained models on HourGlass[10,20] and HourGlass[31,40] to check if they can generalize well on smaller and larger graphs. The experiment results are shown in Table 1. We can see that this dataset can be easily classified by our model based on 2-parameter persistence modules with good generalization performance. But it is not easy for 1-parameter persistence method like PersImg or some GNN models.

| Testing accuracy of models on HourGlass | | | | |
|---|---|---|---|---|
| **Model** | **GCN** | **GIN** | **PersImg** | **PERSGRIL** |
| HourGlass[21,30] | 87.25±4.0 | 84.00±4.4 | 74.00±7.4 | 100.0±0.0 |
| HourGlass[10,20] | 67.31±4.6 | 62.98±3.4 | 50.33±1.6 | 99.79±0.1 |
| HourGlass[31,40] | 87.75±2.2 | 79.10±6.2 | 86.95±5.0 | 100.0±0.0 |

Table 1: Table of testing results from different models. Last two rows show the testing results on HourGlass[10,20] and HourGlass[31,40] of models trained on HourGlass[21,30].

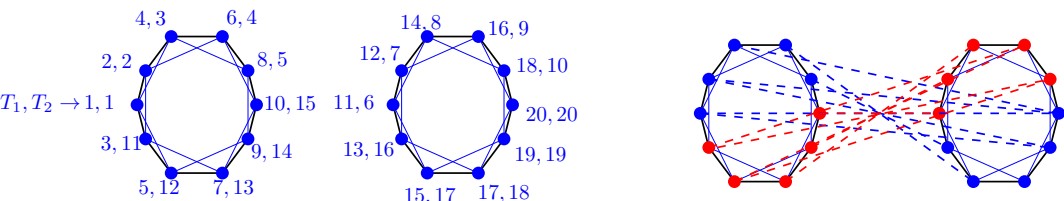

Figure 5: (Left) An example of a graph consisting of two circulant subgraphs. The pair of indices on each node represents the its order on the traversals $T_1$ and $T_2$ respectively. Both traversals start from the left node as the root node. (Right) Cross edges placed across two subgraphs.

**Graph experiments.** We perform a series of experiments on graph classification to test the proposed model. We use standard datasets such as PROTEINS, DHFR, COX2 and MUTAG (Morris et al., 2020). A quantitative summary of these datasets is given in Appendix D.2. On these datasets, we compare the performance of GRIL with other models such as multiparameter persistence landscapes (MP-L) (Vipond, 2020), multiparameter persistence images (MP-I) (Carrière & Blumberg, 2020), multiparameter persistence kernel (MP-K) (Corbet et al., 2019) and PersLay (Carrière et al., 2020).

In (Carrière & Blumberg, 2020), the authors use the heat kernel signature (HKS) and Ricci curvature on the graphs to form a bi-filtration. We also use the same bi-filtration and report the result in the column **Gril HKS-RC**. Since the graphs in all of these datasets have node attributes, we also form Density-Alpha bi-filtration on the node features and compare the performance of GRIL on this bi-filtration (**Gril D-Alpha**) with other methods. Density-Alpha bi-filtration uses Distance-to-Measure function as the filtration function for one coordinate and an Alpha complex filtration in the other coordinate. We use a simple 1-layer MLP as a classifier in order to test the discriminating power of GRIL features. We can see from Table 2 that GRIL with 1-layer MLP gives better performance as compared to multiparameter persistence image, multiparameter persistence kernel, multiparameter

persistence landscapes on PROTEINS, COX2 and MUTAG. However it doesn't seem to perform as well on DHFR. On PROTEINS, COX2 and MUTAG, GRIL has comparable performance with PersLay. (Carrière et al., 2020). Perslay also uses a 1-layer MLP as the classifier. However, PersLay uses spectral features of the graph along with the 1-parameter persistence diagrams corresponding to the filtration given by the heat kernel signature as the filtration function. The reported results are after ten-fold cross validation. The full details of the experiments are given in Appendix D.2

| Dataset | MP-I | MP-K | MP-L | PersLay | GRIL HKS-RC | GRIL D-Alpha |
|---------|------|------|------|---------|-------------|--------------|
| PROTEINS | $67.3 \pm 3.5$ | $67.5 \pm 3.1$ | $65.8 \pm 3.3$ | $74.8 \pm 0.3$ | $71.6 \pm 4.2$ | $72.6 \pm 4.9$ |
| DHFR | $80.2 \pm 2.3$ | $81.7 \pm 1.9$ | $79.5 \pm 2.3$ | $80.3 \pm 0.8$ | $71.3 \pm 3.7$ | $61.8 \pm 2.5$ |
| COX2 | $77.9 \pm 2.7$ | $79.9 \pm 1.8$ | $79.0 \pm 3.3$ | $80.9 \pm 1.0$ | $78.7 \pm 0.0$ | $80.6 \pm 2.5$ |
| MUTAG | $85.6 \pm 7.3$ | $86.2 \pm 2.6$ | $85.7 \pm 2.5$ | $89.8 \pm 0.9$ | $89.4 \pm 7.0$ | $88.4 \pm 8.4$ |

Table 2: 10-fold cross validation test accuracy of different models on graph datasets. The values of the MP-I, MP-K, MP-L columns are as reported in (Carrière & Blumberg, 2020) and those in the PersLay column are as reported in (Carrière et al., 2020).

We have compared the performance of this model with different values of $k$ in $\lambda(\boldsymbol{p}, k, \ell)$. The results are reported in Table 5 in Appendix D. We report the computation times for these datasets in Table 6 in Appendix D. In Table 4, we show the performance of GRIL with different grid resolutions.

**Differentiability of PERSGRIL: A proof of concept.** In the previous experiments we showed how PERSGRIL can be used to obtain topological signatures from graphs to facilitate a specific downstream task, which in our case is a graph classification problem. In that application, we build PERSGRIL on a static filtration function. By static, we mean that we computed the topological features and used them as an input to a classifier. In this experiment, we demonstrate, as a *proof of concept*, how PERSGRIL can be easily integrated in a differentiable framework (with the theoretical foundation laid in Sec.2, specifically Theorem 2.5) like the standard neural network architectures. We show this by rearranging the positions of input points, i.e. encouraging formation of clusters, holes by choosing suitable loss functions.

As shown in Figure 6, input to PERSGRIL is points sampled non-uniformly from two circles. Recall that GRIL is defined over a 2-parameter persistence module induced by some filtration function $f = (f_x, f_y)$. For every vertex $v$, we assign $f_x(v) = 1 - \exp(\frac{1}{\alpha} \sum_{i=1}^{\alpha} d(v, v_i))$, where $v_i$ denotes $i$-th nearest neighbor of the vertex $v$ and $d(v, v_i)$ denotes the distance between $v$ and $v_i$. For our experiments we fix $\alpha = 5$. We set $f_y(v) = 0$. We compute ALPHACOMPLEX filtration (Edelsbrunner & Harer, 2010) of the points and for each edge $e := (u, v)$ we assign $f_x(e) = \max(f_x(u), f_x(v))$ and $f_y(e) = 1 - \exp(d(u, v))$. To obtain a valid bi-filtration function on the simplicial complex we extend the bi-filtration function from 1-simplices to 2-simplices, i.e. triangles. We pass $f$ as an input to PERSGRIL, coded with the framework PYTORCH (Paszke et al., 2019), that computes persistence landscapes. PERSGRIL uniformly samples $n$ center points from the grid $[0, 1]^2$. Since GRIL value computation can be done independently for each $k$ and a center point, we take advantage of parallel computation and implement the code in a parallel manner. In the forward pass we get GRIL values $\lambda(\boldsymbol{p}, k, \ell)$ for generalized rank $k = 1, 2$, worm size $\ell = 2$ and homology of dimension 1 while varying $\boldsymbol{p}$ over all the sampled center points. After we get the GRIL values, we compute the assignment $s$ according to Theorem 2.5. During the backward pass, we utilize this assignment to compute the derivative of PERSGRIL with respect to the filtration function and consequently update it. We get $n$ values of $\lambda(\cdot, 1, 2)$ for $n$ center points. We treat these $n$ values as a vector and denote is as $\lambda_1$. Similarly, we use $\lambda_2$ to denote the vector formed by values $\lambda(\cdot, 2, 2)$. We minimize the loss $L = -(\|\lambda_1\|_2^2 + \|\lambda_2\|_2^2)$. Figure 6 shows the result after running PERSGRIL for 200 epochs. The optimizer we use to optimize the loss function is Adam (Kingma & Ba, 2015) with a learning rate of 0.01.

## 5 CONCLUSION

This work presents GRIL, a 2-parameter persistence vectorization based on generalized rank invariant which we show is stable and differentiable with respect to the bi-filtration functions. Further, we present an algorithm for computing GRIL which is a synergistic confluence of the recent developments in computing generalized rank invariant of a 2-parameter module and an efficient algorithm

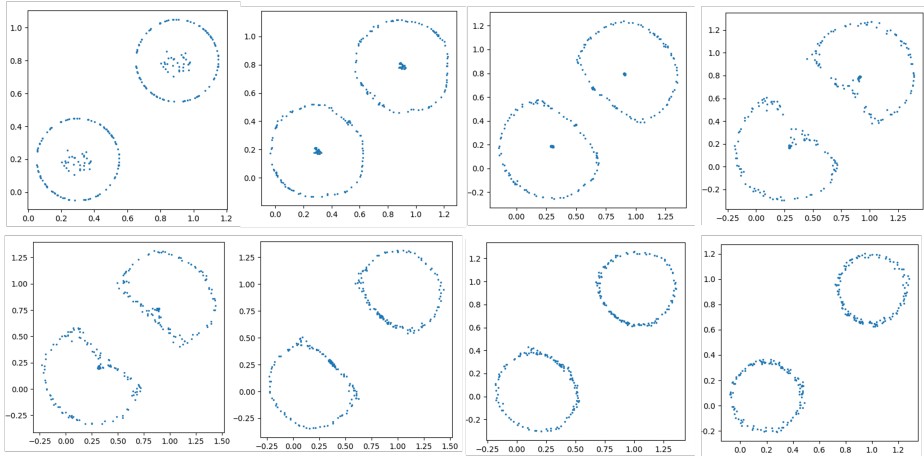

Figure 6: The figures show the rearrangement of points according to the loss function, which in our case is increasing the norm of $\lambda_1$ and $\lambda_2$ vectors. We start with two circles containing some noisy points inside. We observe that the points rearrange to form two circles because that increases the norm of $\lambda_1$ and $\lambda_2$ vectors.

for computing zigzag persistence. We propose PERSGRIL, a differentiable topological layer, which can be used as a topological feature extractor in a differentiable manner. As a topological feature extractor, PERSGRIL can perform better than Graph Convolutional Networks (GCNs) and Graph Isomorphism Networks (GINs) on some synthetic datasets. It performs better than the existing multiparameter persistence methods on some graph benchmark datasets. Further, we give a proof of concept for the differentiability of PERSGRIL by rearranging the point cloud to enhance its topological features. We believe that the additional topological information that a 2-parameter persistence module encodes, as compared to a 1-parameter persistence module, can be leveraged to learn better representations. Further directions of research include using PERSGRIL with GNNs for filtration learning to learn more powerful representations. We hope that this work motivates further research into exploring this direction.

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

## A BACKGROUND AND DEFINITIONS

Here, we give the detailed definitions of all the concepts explained in the paper. We begin by defining a simplicial complex.

**Definition A.1** (Simplicial Complex). An abstract simplicial complex is a pair $(V, \Sigma)$ where $V$ is a finite set and $\Sigma$ is a collection of non-empty subsets of $V$ such that if $\sigma \in \Sigma$ and if $\tau \subseteq \sigma$ then $\tau \in \Sigma$. A topological space $|(V, \Sigma)|$ can be associated with the simplicial complex which can be defined using a bijection $t \colon V \to \{1, 2, \ldots, |V|\}$ as the subspace of $\mathbb{R}^{|V|}$ formed by the union $\bigcup_{\sigma \in \Sigma} h(\sigma)$, where $h(\sigma)$ denotes the convex hull of the set $\{e_{t(s)}\}_{s \in \sigma}$, where $e_i$ denotes the standard basis vector in $\mathbb{R}^{|V|}$.

We shall now define a zigzag filtration and the zigzag persistence module induced by it.

**Definition A.2.** A *zigzag filtration* is a sequence of simplicial complexes where both insertions and deletions of simplices are allowed, the possibility of which we indicate with double arrows:

$$X_0 \leftrightarrow X_1 \leftrightarrow \cdots \leftrightarrow X_n = \mathcal{X}.$$

Applying homology functor on such a filtration we get a zigzag persistence module that is a sequence of vector spaces connected either by forward or backward linear maps:

$$H_*(X_0) \leftrightarrow H_*(X_1) \leftrightarrow \cdots \leftrightarrow H_*(X_n).$$

Now, we give the definition of 2-parameter filtration over $\mathbb{R}^2$ and the 2-parameter persistence module induced by it.

**Definition A.3** (2-parameter simplicial filtration over $\mathbb{R}^2$). A 2-parameter simplicial filtration, also called bi-filtration, over $\mathbb{R}^2$ is a collection of simplicial complexes $\{X_{\mathbf{u}}\}_{\mathbf{u} \in \mathbb{R}^2}$ with inclusion maps $X_{\mathbf{u}} \hookrightarrow X_{\mathbf{v}}$ for $\mathbf{u} \leq \mathbf{v}$, that is, $u_1 \leq u_2$ and $v_1 \leq v_2$ where $\mathbf{u} = (u_1, u_2)$ and $\mathbf{v} = (v_1, v_2)$.

**Definition A.4** (2-parameter Persistence Module). Given a bi-filtration, $\{X_{\mathbf{u}}\}_{\mathbf{u} \in \mathbb{R}^2}$, by considering the homology of the simplicial complexes in the bi-filtration over the finite field $\mathbb{Z}_2$, we get a collection of vector spaces $\{M_{\mathbf{u}} \mid \mathbf{u} \in \mathbb{R}^2\}$ along with a collection of linear maps $\{M_{\mathbf{u} \to \mathbf{v}} : M_{\mathbf{u}} \to M_{\mathbf{v}} \mid \mathbf{u} \leq \mathbf{v}\}$. Each inclusion map in the bi-filtration induces a linear map between the corresponding homology vector spaces.

Having defined 2-parameter filtration and 2-parameter persistence module, we now define the notion of an Interval in $\mathbb{R}^2$. In the definition, we shall make use of the standard partial order on $\mathbb{R}^2$, i.e., $\mathbf{u} \leq \mathbf{v}$ if $u_1 \leq v_1$ and $u_2 \leq v_2$ for $\mathbf{u} = (u_1, u_2)$ and $\mathbf{v} = (v_1, v_2)$.

**Definition A.5.** An interval in $\mathbb{R}^2$ is a subset $\emptyset \neq I \subseteq \mathbb{R}^2$ that satisfies the following:

1. If $\mathbf{p}, \mathbf{q} \in I$ and $\mathbf{p} \leq \mathbf{r} \leq \mathbf{q}$, then $\mathbf{r} \in I$;

2. If $\mathbf{p}, \mathbf{q} \in I$, then there exists a finite sequence $(\mathbf{p} = \mathbf{p}_0, \mathbf{p}_1, , \ldots, \mathbf{p}_m = \mathbf{q}) \in I$ so that every consecutive points $\mathbf{p}_i, \mathbf{p}_{i+1}$ are comparable in the partial order for $i \in \{0, \ldots, m-1\}$.

We now give the formal definition of generalized rank invariant over intervals in $\mathbb{R}^2$. However, generalized rank invariant can be defined over any locally finite connected poset.

**Definition A.6** (Generalized Rank (Kim & Mémoli, 2021)). Given a 2-parameter persistence module $M$ and an intervals $I \subseteq \mathbb{R}^2$, the *generalized rank* of $M$ restricted to $I$, $\mathrm{rk}^M(I)$, is defined as

$$\mathrm{rk}^M(I) \triangleq \mathrm{rank}[\varprojlim M|_I \to \varinjlim M|_I].$$

Here $\varprojlim M|_I, \varinjlim M|_I$ denote the limit and colimit of the functor $M$ when restricted to $I$. We refer the reader to (MacLane, 1971) for the definitions of limit and colimit in category theory.

For a collection of intervals $\mathcal{I}$, the collection $\mathrm{rk}_{\mathcal{I}}^M \triangleq \{\mathrm{rk}^M(I) \mid I \in \mathcal{I}\}$ is called *generalized rank invariant* of $M$ over $\mathcal{I}$.

We can define a metric on the space of persistence modules based on their generalized rank invariants over all intervals in $\mathbb{R}^2$.

**Definition A.7** (Erosion Distance (Patel, 2018; Kim & Mémoli, 2021)). Let $\mathbf{Int}(\mathbb{R}^2)$ be the collection of all intervals in $\mathbb{R}^2$. Let $M$ and $N$ be two persistence modules. The *erosion distance* is defined as

$$\mathrm{d}_{\mathcal{E}}(M, N) \triangleq \inf_{\varepsilon \geq 0} \{\forall I \in \mathbf{Int}(\mathbb{R}^2), \mathsf{rk}^M(I) \geq \mathsf{rk}^N(I^{+\varepsilon}) \text{ and } \mathsf{rk}^N(I) \geq \mathsf{rk}^M(I^{+\varepsilon})\}.$$

Here $I^{+\varepsilon}$ denotes the $\varepsilon$-extension of the interval $I$.

# B  STABILITY AND DIFFERENTIABILITY: PROOFS

In this section, we provide the proof for stability and differentiability of GRIL. We begin by defining some metrics on the space of persistence modules based on GRIL.

**Definition B.1.** Given two persistence modules $M$ and $N$, a morphism $f : M \to N$ is a collection of linear maps $\{f_{\mathbf{u}} : M_{\mathbf{u}} \to N_{\mathbf{u}}\}_{\mathbf{u} \in \mathbb{R}^2}$ such that $f_{\mathbf{u}} \circ N_{\mathbf{u} \to \mathbf{v}} = M_{\mathbf{u} \to \mathbf{v}} \circ f_{\mathbf{v}}, \forall \mathbf{u} \leq \mathbf{v}$.

**Definition B.2.** Given a persistence module $M$ and $\epsilon \in \mathbb{R}$, we define the *shift module* $M^{\leftarrow \epsilon}$ through $M_{\mathbf{u}}^{\leftarrow \epsilon} = M_{\mathbf{u}+\epsilon}$ and $M_{\mathbf{u} \to \mathbf{v}}^{\leftarrow \epsilon} = M_{\mathbf{u}+\epsilon \to \mathbf{v}+\epsilon}$. Here $\mathbf{u} + \epsilon = (\mathbf{u}_1 + \epsilon, \mathbf{u}_2 + \epsilon)$.

**Definition B.3.** For a pair of persistence module $M$ and $N$ and some $\epsilon \in \mathbb{R}_{\geq 0}$, an $\epsilon$-interleaving between $M$ and $N$ is a pair of morphisms $\phi : M \to N^{\leftarrow \epsilon}$ and $\psi : N \to M^{\leftarrow \epsilon}$ such that $\forall \mathbf{u} \in \mathbb{R}^2, M_{\mathbf{u} \to \mathbf{u}+2\epsilon} = \psi_{\mathbf{u}+\epsilon} \circ \phi_{\mathbf{u}}$ and $N_{\mathbf{u} \to \mathbf{u}+2\epsilon} = \phi_{\mathbf{u}+\epsilon} \circ \psi_{\mathbf{u}}$. If such interleaving exists, we say $M$ and $N$ are $\epsilon$-interleaved.

**Definition B.4.** For two persistence modules $M$ and $N$, we define the *interleaving distance* as $\mathrm{d}_{\mathrm{I}}(M, N) \triangleq \inf_{\epsilon \geq 0}\{M \text{ and } N \text{ are } \epsilon\text{-interleaved}\}$.

**Definition B.5.** For persistence module $M, N$ with GRILs $\lambda^M, \lambda^N$, define

$$d_{\mathcal{L}}(M, N) \triangleq ||\lambda^M - \lambda^N||_{\infty}.$$

We shall now look at a property of GRIL that will help in proving the stability.

**Definition B.6.** Given any interval $I$ and $\varepsilon \geq 0$, let $I^{+\varepsilon}$ be the $\varepsilon$-*extension* of $I$ defined as:

$$I^{+\varepsilon} \triangleq \bigcup_{\mathbf{p} \in I} \boxed{\mathbf{p}}_{\varepsilon} \tag{2}$$

where $\boxed{\mathbf{p}}_{\varepsilon} \triangleq \{\mathbf{q} : ||\boldsymbol{p} - \boldsymbol{q}||_{\infty} \leq \varepsilon\}$ is the $\infty$-norm $\varepsilon$-neighbourhood of $x$.

**Proposition B.1.** $\left(\boxed{\boldsymbol{p}}_{\delta}^{\ell}\right)^{+\varepsilon} \subseteq \boxed{\boldsymbol{p}}_{\delta+\varepsilon}^{\ell}$.

In order to better analyze the stability property of persistence landscape, we define a distance in a similar flavour as erosion distance for the underlying collection of all worms.

**Notation B.7.** Denote the collection of all worms as $\mathcal{W} \triangleq \left\{\boxed{\boldsymbol{p}}_{\delta}^{\ell} \mid \delta > 0, l \in \mathbb{N}_+, \boldsymbol{p} \in \mathbb{R}^2\right\}$.

**Definition B.8.** For $\mathcal{W} \triangleq \left\{\boxed{\boldsymbol{p}}_{\delta}^{\ell} \mid \delta > 0, l \in \mathbb{N}_+, \boldsymbol{p} \in \mathbb{R}^2\right\}$, define a distance $\mathrm{d}_{\mathcal{E}}^{\mathcal{W}}$ as follows:

$$\mathrm{d}_{\mathcal{E}}^{\mathcal{W}}(M, N) \triangleq \inf \left\{\varepsilon \mid \forall \boxed{\boldsymbol{p}}_{\delta}^{\ell} \in \mathcal{W}, [\mathsf{rk}^M\left(\boxed{\boldsymbol{p}}_{\delta}^{\ell}\right) \geq \mathsf{rk}^N\left(\boxed{\boldsymbol{p}}_{\varepsilon+\delta}^{\ell}\right) \text{ and } \mathsf{rk}^N\left(\boxed{\boldsymbol{p}}_{\delta}^{l}\right) \geq \mathsf{rk}^M\left(\boxed{\boldsymbol{p}}_{\varepsilon+\delta}^{\ell}\right)]\right\}. \tag{3}$$

**Proposition B.2.** $\mathrm{d}_{\mathcal{L}} = \mathrm{d}_{\mathcal{E}}^{\mathcal{W}} \leq \mathrm{d}_{\mathcal{E}}$, *where $\mathrm{d}_{\mathcal{E}}$ is the erosion distance.*

*Proof.* $\mathrm{d}_{\mathcal{E}}^{\mathcal{W}} \leq \mathrm{d}_{\mathcal{E}}$ is obvious by definition.

To show $\mathrm{d}_{\mathcal{L}} \leq \mathrm{d}_{\mathcal{E}}^{\mathcal{W}}$. Given two persistence modules $M, N$, assume $\mathrm{d}_{\mathcal{E}}^{\mathcal{I}}(M, N) = \epsilon$. For fixed $\boldsymbol{p}, k, \ell$, let $\lambda^M(\boldsymbol{p}, k, \ell) = \delta_1$ and $\lambda^N(\boldsymbol{p}, k, \ell) = \delta_2$. Without loss of generality, assume $\delta_2 \geq \delta_1$. We want to show that $\delta_2 - \delta_1 \leq \epsilon$. By the construction of $\mathrm{d}_{\mathcal{E}}^{\mathcal{W}}$, we know that for any $\alpha > 0$, $k > \mathsf{rk}^N(\boxed{\boldsymbol{p}}_{\delta_1+\alpha}^{\ell}(x)) \geq \mathsf{rk}^M(\boxed{\boldsymbol{p}}_{\delta_1+\epsilon+\alpha}^{\ell}(x))$. One can get $\delta_1 + \epsilon + \alpha > \delta_2 \implies \epsilon + \alpha > \delta_2 - \delta_1$. By taking $\alpha \to 0$, we have $\delta_2 - \delta_1 \leq \epsilon$.

To show $d_{\mathcal{E}}^{\mathcal{W}} \leq d_{\mathcal{L}}$. Let $d_{\mathcal{L}}(M, N) = \delta$. For any $I = \boxed{\boldsymbol{p}}_{\epsilon}^{\ell} \in \mathcal{I}$, we want to show that $\mathsf{rk}^M(\boxed{\boldsymbol{p}}_{\epsilon}^{\ell}) \geq \mathsf{rk}^N(\boxed{\boldsymbol{p}}_{\epsilon+\delta}^{\ell})$ and $\mathsf{rk}^N(\boxed{\boldsymbol{p}}_{\epsilon}^{\ell}) \geq \mathsf{rk}^M(\boxed{\boldsymbol{p}}_{\epsilon+\delta}^{\ell})$. We prove the first inequality. The second one can be proved in a similar way. Let $k = \mathsf{rk}^N(\boxed{\boldsymbol{p}}_{\epsilon+\delta}^{\ell})$, then $\lambda^N(\boldsymbol{p}, k, \ell) \geq \epsilon + \delta$. By the assumption $d_{\mathcal{L}}(M, N) = \delta$, we know that $\lambda^N(\boldsymbol{p}, k, \ell) \geq \epsilon$, which implies $\mathsf{rk}^M(\boxed{\boldsymbol{p}}_{\epsilon}^{\ell}) \geq k = \mathsf{rk}^N(\boxed{\boldsymbol{p}}_{\epsilon+\delta}^{\ell})$. $\quad\square$

**Proposition.** 2.1 GRIL is equivalent to the generalized rank invariant on $\mathcal{W}$. Here equivalence means bijective reconstruction from each other.

*Proof.* Constructing GRIL from generalized rank invariant on $\mathcal{W}$ is immediate from the definition of GRIL.

On the other direction, for any $\boldsymbol{p}, \delta, \ell$, the generalized rank $\mathsf{rk}_{\mathcal{W}}^M(\boxed{\boldsymbol{p}}_{\delta}^{\ell})$ can be reconstructed by GRIL as follows:

$$\mathsf{rk}_{\mathcal{W}}^M(\boxed{\boldsymbol{p}}_{\delta}^{\ell}) = \arg\max_k \{\lambda(\boldsymbol{p}, k, \ell) \geq \delta\} \tag{4}$$

It is not hard to check that, this construction, combined with the construction of persistence landscape, gives a bijective mapping between (generalized) rank invariants over $\mathcal{W}$ and GRILs. $\quad\square$

By the stability property of erosion distances, we can immediately get the stability of GRIL as follows:

**Proposition.** 2.2 For two filtration functions $f, f' \colon \mathcal{X} \to \mathbb{R}^2$, $||\lambda^{M^f} - \lambda^{M^{f'}}||_{\infty} \leq ||f - f'||_{\infty}$.

*Proof.* Let $M^f$ and $M^{f'}$ be the persistence modules derived by $f$ and $f'$ respectively. Then, we have the following chain of inequalities:

$$\|\lambda^{M^f} - \lambda^{M^{f'}}\|_{\infty} = d_{\mathcal{L}}(M^f, M^{f'}) \leq d_{\mathcal{E}}(M^f, M^{f'}) \leq d_I(M^f, M^{f'}) \leq \|f - f'\|_{\infty}$$

where $d_I(M^f, M^{f'})$ is the interleaving distance. The second last inequality has been shown in (Kim & Mémoli, 2021). $\quad\square$

Recall that when $\mathcal{X}$ is a finite space (e.g. finite simplicial complex, point cloud) then, any $f \colon \mathcal{X} \to \mathbb{R}^2$ can be considered as an $n \times 2$ matrix which can be linearized into a vector in $\mathbb{R}^{2n}$. Let us denote that vector by $v_f$.

**Proposition** (2.3). PERSGRIL is Lipschitz continuous with respect to bi-filtration functions on finite spaces.

*Proof.* Given filtration functions $f, f'$ and their corresponding vector representations $v_f, v_{f'} \in \mathbb{R}^{2n}$, it is easy to see that $\|f - f'\|_{\infty} \leq 2\|v_f - v_{f'}\|_{\infty} \leq 2\|v_f - v_{f'}\|$. Combining this with the chain of inequalities in the previous proposition, we get that PERSGRIL is Lipschitz continuous with respect to the underlying filtration functions. $\quad\square$

**Theorem** (2.5). Consider the space of all filtration functions $\{f : \mathcal{X} \to \mathbb{R}^2\}$ on a finite space $\mathcal{X}$ with $|\mathcal{X}| = n$, which is equivalent to $\mathbb{R}^{2n}$. For fixed $k, \ell, \boldsymbol{p}$, there exists a measure-zero subset $Z \subseteq \mathbb{R}^{2n}$ such that for any $f \in \mathbb{R}^{2n} \setminus Z$ satisfying the following generic condition: $\forall x \neq y \in \mathcal{X}, f(x)_1 \neq f(y)_1, f(x)_2 \neq f(y)_2$, there exists an assignment $s : \mathcal{X} \to \{\pm 1, 0, \pm \ell\}^2$ such that

$$\nabla_s \Lambda_{\boldsymbol{p}}^{k,\ell}(f) \triangleq \lim_{\alpha \to 0} \frac{\Lambda_{\boldsymbol{p}}^{k,\ell}(f + \alpha s) - \Lambda_{\boldsymbol{p}}^{k,\ell}(f)}{\alpha \|s\|_{\infty}} = \max_{g \in \mathcal{X}} \nabla_g \Lambda_{\boldsymbol{p}}^{k,\ell}(f).$$

*Proof.* By Corollary 2.4 we know there exists some measure-zero set $R \subset \mathbb{R}^{2n}$ such that PERSGRIL is differentiable in $\bar{R} \triangleq \mathbb{R}^{2n} \setminus R$. Let $M = M^f$ be a 2-parameter persistence module induced from some generic filtration function $f \in \bar{R}$ and $I = \boxed{\boldsymbol{p}}_{d}^{\ell}$ be an $\ell$-worm in $\mathbb{R}^2$ centered at some point $\boldsymbol{p}$. Let $\partial(I)$ be the boundary of $I$ excluding the right most vertical edge and bottom most vertical edge (See Figure 7 as an illustration). It is shown in (Dey et al., 2022) that, over the boundary $\partial(I)$, a so-called zigzag persistence module can be defined by taking the restricting $M$ to $\partial(I)$ (in practice

it is enough to take a zigzag path to approximate the smooth off-diagonal boundary) on which the number of full bars is equal to $\mathsf{rk}^M(I)$. Let $I' = \boxed{\boldsymbol{p}}_{d'}^{\ell}$ be another $\ell$-worm centered at $\boldsymbol{p}$ for some $d' \neq d$. One can observe that, if the zigzag filtrations on $\partial(I)$ and $\partial(I')$ have the same order of insertion and deletion of simplices , then the number of full bars on $M|_{\partial(I)}$ and $M|_{\partial(I')}$ are the same, which means $\mathsf{rk}^M(I) = \mathsf{rk}^M(I')$. Now let $d = \lambda^M(k, \ell, \boldsymbol{p}), I = \boxed{\boldsymbol{p}}_d^{\ell}, I_- = \boxed{\boldsymbol{p}}_{d-\epsilon}^{\ell}, I_+ = \boxed{\boldsymbol{p}}_{d+\epsilon}^{\ell}$ for some small enough $\epsilon$. Based on the definition of $\lambda^M$, we know that $\mathsf{rk}^M(I_-) \geq k$ and $\mathsf{rk}^M(I_+) < k$, which means that zigzag filtrations change on some simplices while moving from $\partial(I_-)$ to $\partial(I_+)$. Either the collection of simplices changes or the order of simplices changes. The former case corresponds to the simplices with $x$ or $y$-coordinate aligned with some vertical or horizontal edges on the $\partial(I)$. The latter case corresponds to those pairs of simplices $(\sigma, \tau)$ such that $f(\sigma) \vee f(\tau) \triangleq (\max(f(\sigma)_1, f(\tau)_1), \max(f(\sigma)_2, f(\tau)_2)$ is on some off-diagonal edges on $\partial(I)$. By the generic condition of the filtration function $f$, we can locate those simplices as the set $S$, which we call support simplices. The assignment function $s$ is defined on each $\sigma \in S$ by assigning $s(\sigma) = \pm 1$ or $\pm \ell$ which is consistent with the moving direction of the edge from $\partial(I)$ to $\partial(I_+)$. We discuss the assignment values case by case:

We can divide the boundary into four edges: bottom (off-diagonal) edge $e_b$, top (horizontal) edge $e_t$, left (vertical) edge $e_l$, right (off-diagonal) edge $e_r$.

1. $s(\sigma) = (0, +\ell)$ if $\sigma$ has $y$-coordinate the same as $e_t$,

2. $s(\sigma) = (-\ell, 0)$ if $\sigma$ has $x$-coordinate the same as $e_l$,

3. $s(\sigma) = (0, -1), s(\tau) = (-1, 0)$ if $f(\sigma) \vee f(\tau)$ is on $e_b$ and $f(\sigma)_1 \leq f(\tau)_1$,

4. $s(\sigma) = (0, +1), s(\tau) = (+1, 0)$ if $f(\sigma) \vee f(\tau)$ is on $e_r$ and $f(\sigma)_1 \leq f(\tau)_1$,

See Figure 7 as an illustration. We assume $f$ satisfies the condition that the supporting simplices in $S$ either all belong to cases 1 and 2 or all belong to cases 3 and 4, but not a combination of them. It is not hard to see that the collection of $f$ for which this condition does not hold is a measure zero set in $\mathbb{R}^{2n}$. Let us denote the collection of all such $f$'s by $F$. Then, $Z = F \cup R$ is a measure zero set in $\mathbb{R}^{2n}$ which consists of $f$'s which do not satisfy the condition and those points where PERSGRIL is not differentiable.

Now, check for such a generic $f \notin Z$ so that the directional derivative $\nabla_s \lambda(f)$ is indeed a maximal directional derivative. For the cases 3 and 4, the stability property in Proposition 2.2 implies that, for any $\alpha > 0$ and any direction vector $g \in \mathbb{R}^{2n}$ with $\|g\|_\infty = 1$, we have $\lambda(f + \alpha g) - \lambda(f) \leq \alpha$. Also it is not hard to check that $\lambda(f + \alpha s) - \lambda(f) = \alpha$ for $\alpha > 0$ small enough since the zigzag persistence of $M^{f+\alpha s}|_J$ with $J = \boxed{\boldsymbol{p}}_{d+\alpha}^{\ell}$ has the same collection of simplices and orders as $M^f|_I$ with $I = \boxed{\boldsymbol{p}}_d^{\ell}$, which means they have the same rank. Therefore, we have $\forall \|g\|_\infty = 1, \lambda(f + \alpha g) - \lambda(f) \leq \lambda(f + \alpha s) - \lambda(f) \implies \nabla_g \Lambda(f) \leq \nabla_s \Lambda(f)$. For the case 1 (the case 2 is similar), the support simplex is on edge $e_t$. Now for any direction vector $g \in \mathbb{R}^{2n}$ and $\alpha > 0$ small enough, let $\Delta d = \Lambda(f + \alpha g) - \Lambda(f)$ and let $\Delta y_{e_t}$ be the difference between $y$-coordinates of $e_t$'s from $\boxed{\boldsymbol{p}}_d^{\ell}$ and $\boxed{\boldsymbol{p}}_{d+\Delta d}^{\ell}$. Note that $\frac{\Delta d}{\Delta y_{e_t}} = \ell$ and $\frac{|\Lambda(f+\alpha g)-\Lambda(f)|}{\alpha \|g\|_\infty} \leq \frac{\Delta d}{\Delta y_{e_t}}$ since in order to change $\Lambda(f)$ by $\Delta d$ one has to at least move edge $e_t$ by $\Delta y_{e_t}$, which correspondingly changes the $y$-coordinate of $s(\sigma)$ by $\Delta y_{e_t}$. From the above argument, we can get the directional derivative $\nabla_g \Lambda(f)$ is bounded from above by the ratio $\frac{\Delta d}{\Delta y_{e_t}} = \frac{1}{\ell} = \nabla_s \Lambda(f)$. The case for $\alpha < 0$ is symmetric.

In summary, $\nabla_s \lambda(f)$ indeed maximizes the directional derivative for $f$. $\qquad \square$

## C  ALGORITHM

Here, we describe the algorithm in detail. In practice, we are usually presented with a piecewise linear (PL) approximation $\hat{f}$ of a $\mathbb{R}^2$-valued function $f$ on a discretized domain such as a finite simplicial complex. The PL-approximation $\hat{f}$ itself is $\mathbb{R}^2$-valued. Discretizing the parameter space

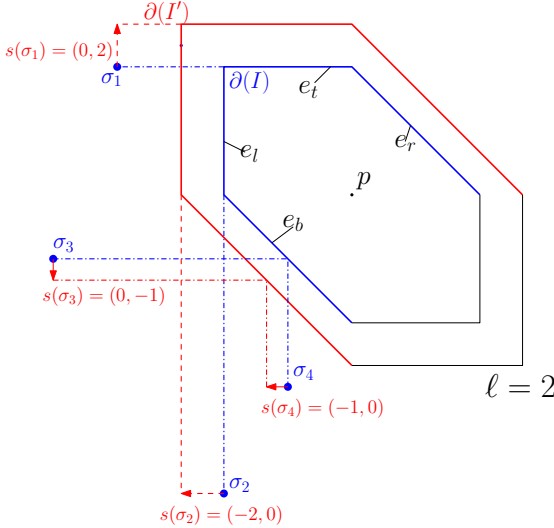

Figure 7: Two examples of 2-worm $I, I'$. Blue and red lines are boundaries of $I$ and $I'$ respectively on which the zigzag persistence modules are constructed for computing ranks. $\sigma_i, i = 1, 2, 3, 4$ are four support simplices on $\partial(I)$. $s(\sigma_i)$ is the assignment function values on $\sigma_i$.

$\mathbb{R}^2$ by a grid, we consider a *lower star* bi-filtration of the simplicial complex. Analogous to the 1-parameter case, a lower star bi-filtration is obtained by assigning every simplex the maximum of the values over all of its vertices in each of the two co-ordinates. With appropriate scaling, these (finite) values can be mapped to a subset of points in a uniform finite grid over $[0, 1] \times [0, 1]$. Observe that because of the maximization of values over all vertices, we have the property that two simplices $\sigma \subseteq \tau$ have values $\hat{f}(\sigma) \in \mathbb{R}^2$ and $\hat{f}(\tau) \in \mathbb{R}^2$ where $\hat{f}(\sigma) \le \hat{f}(\tau)$. A partial order of the simplices according to these values provide a bi-filtration over the grid $[0, 1] \times [0, 1]$.

**Choosing center points for worms.** Let us denote the chosen grid as $\mathrm{Grid} = \{\left(\frac{m}{M}, \frac{n}{M}\right) \mid m, n \in \{0, 1, \ldots, M\}\}$ for some $M \in \mathbb{Z}_+$. We denote the grid resolution as $\rho \triangleq 1/M$. We sample a uniform subgrid $\mathcal{P} \subseteq \mathrm{Grid}$ as the collection of center points for the worms to be used to build GRIL.

**Discretized $\ell$-worms.** We saw the definition of $\ell$-worm in the previous section. However, in practice, since we work with a discrete grid rather than $\mathbb{R}^2$, we use *discretized $\ell$-worms* as an approximation. The approximation gap is determined by the grid resolution $\rho$. A discretized $\ell$-worm centered at $\boldsymbol{p}$ with width $d$ is the union of $2\ell - 1$ squares with centers at $\boldsymbol{p} + (kd, -kd)$ and $\boldsymbol{p} - (kd, -kd)$ where $k \in \{0, 1, \ldots, \ell - 1\}$ along with the intermediate staircases between two squares of step-size equal to *grid resolution* ($\rho$). Figure 2 (middle) shows the discretization of a 2-worm. This construction is sensitive to the grid resolution.

**Computing generalized ranks.** We need to compute the generalized rank $\mathrm{rk}^M(\boxed{\boldsymbol{p}}_d^\ell)$ for every worm $\boxed{\boldsymbol{p}}_d^\ell$ to decide whether to increase its width or not. We use a result of (Dey et al., 2022) to compute $\mathrm{rk}^M(\boxed{\boldsymbol{p}}_d^\ell)$. It says that $\mathrm{rk}^M(\boxed{\boldsymbol{p}}_d^\ell)$ can be computed by considering a zigzag module and computing the number of full bars (bars that begin at the start of the zigzag filtration and persist until the end of the filtration) in its decomposition. This zigzag module decomposition can be obtained by restricting the bi-filtration on the boundary of $\mathrm{rk}^M(\boxed{\boldsymbol{p}}_d^\ell)$ and using any of the zigzag persistence algorithms on the resulting zigzag filtration. We use the recently published efficient algorithm and its associated software (Dey & Hou, 2022) for computing zigzag persistence.

**Computing the value of GRIL using binary search.** For a worm $\boxed{\boldsymbol{p}}_d^\ell$ and a given $k \ge 1$, we apply binary search to compute the value of GRIL. Let us denote the grid resolution by $\rho$. We do the binary search for $d$ in the range $[d_{\min}, d_{\max}]$ where $d_{\min} = \rho$ and $d_{\max} = length\ of\ the\ grid$. In

each iteration, we compute $\mathsf{rk}^M(\boxed{\boldsymbol{p}}_d^\ell)$ for $d = (d_{\min} + d_{\max})/2$ and check if $\mathsf{rk}^M(\boxed{\boldsymbol{p}}_d^\ell) \geq k$. We increase the width of the worm by updating

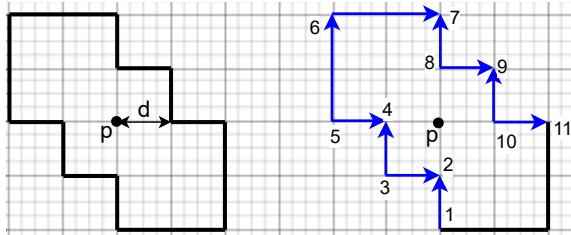

Figure 8: (Left) The figure shows the 2-worm centered at $p$ with width $d$. (Right) The highlighted part denotes the boundary cap of the worm. The arrows in the figure denote the direction of arrows in the zigzag filtration.

Refer to Figure 8 for an illustration of the zigzag filtration along the boundary cap of a 2-worm.

## D EXPERIMENTAL SETUP

### D.1 HOURGLASS DATASET

The two traversals $T_1$ and $T_2$ are designed as follows: $T_1$ traverses $G_1$, then followed by $G_2$; $T_2$ traverses upper halves $G_1^\top \subseteq G_1$ and $G_2^\top \subseteq G_2$ sequentially first, then followed by the other halves $G_1^\perp \subseteq G_1$ and $G_2^\perp \subseteq G_2$. For cross edges, we randomly pick $2|V|$ pairs of nodes (with replacement) in $G_1^\top \times G_2^\perp$ on which we place cross edges. We don't place multiple edges on the same pair of nodes. In a similar way we place cross edges on $G_1^\perp \times G_2^\top$. Therefore, $G$ has roughly $6|V|$ cross edges between $G_1$ and $G_2$. The (roughly) total number of edges: $|E| \approx 5|V|$. For methods based on persistence modules, we take two filtration functions $f_1, f_2 : V \cup E \to \mathbb{R}$ on $G$ as follows: let $x(v)$ be the node attribute on $v$ given by the order index of the trace. Then

- $f_1$ is given by $\forall v \in V, f_1(v) = x(v)$ and $\forall e = (v, w) \in E, f_1(e) = \max(x(v), x(w))$.
- $f_2$ is given by $f_2(v) = 0$ and $f_2 = C(e)$ where $C(e)$ is a curvature value of $e$. Here we use a version of discrete Ricci called Forman-Ricci curvature (Forman, 2003) computed by the code provided in (Ni et al., 2019).

We compute for all points $\boldsymbol{p}$ in a uniform $4 \times 4$ grid the PERSGRIL values $\lambda(\boldsymbol{p}, k, \ell)$ for generalized rank $k = 1, 2$, worm size $\ell = 2$, and homology of dimension 0 and 1. Therefore, for each graph our PERSGRIL generates a 64-dimensional vector as representation. For the method based on 1-parameter persistence modules with persistence image vectorization, we compute 1-parameter persistence modules for homology dimension $0, 1$ on $f_1$ and $f_2$ independently. Each persistence module will be vectorized on a $4 \times 4$ grid. Therefore, it also produces a 64-dimensional vector as representation. For graph neural networks models, we use 3-layer GCN and 3-layer GNN with fixed hidden dimension to be 16, followed by sum pooling and one fully-connected layer. We use 3-layer multilayer perceptron (MLP) with fixed hidden dimension to be 16 as classifiers for all models. We train all the models 100 epochs with cross entropy loss and Adam optimizer (Kingma & Ba, 2015) with learning rate fixed to be lr=0.001. We do 5-fold cross validation and report the mean accuracy and standard deviation.

### D.2 GRAPH EXPERIMENTS

We performed a series of experiments on graph classification using GRIL. We used standard datasets with node features such as PROTEINS, DHFR, COX2 and MUTAG (Morris et al., 2020). Description of the graph classification tasks is given in Table 3. The node features of all the nodes were treated as points in a higher dimensional space and we computed the Density-Alpha bi-filtration on the nodes. We extended the filtration on the edges by considering the maximum of the values on the corresponding nodes.

| Dataset | Num Graphs | Num Classes | Avg. No. Nodes | Avg. No. Edges |
|---|---|---|---|---|
| PROTEINS | 1113 | 2 | 39.06 | 72.82 |
| COX2 | 467 | 2 | 41.22 | 43.45 |
| DHFR | 756 | 2 | 42.43 | 44.54 |
| MUTAG | 188 | 2 | 17.93 | 19.79 |

Table 3: Description of Graph Datasets

The Density-Alpha bi-filtration and the Heat Kernel Signature-Ricci Curvature bi-filtration, as done in (Carrière & Blumberg, 2020), values are normalized so that they lie between 0 and 1. For the experiments reported in 4, we fix the grid resolution $\rho = 0.01$. Thus, the square $[0, 1] \times [0, 1]$ has $100 \times 100$ many grid points. We sample 128 center points, $\boldsymbol{p}$, out of these grid points uniformly. We fix $l = 2$ for our experiments. We compute $\lambda(\boldsymbol{p}, k, \ell)$ where $\boldsymbol{p}$ varies over the sampled 128 center points and $k$ varies from 1 to 10. Each such computation is done for dimension 0 homology ($H_0$) and dimension 1 homology ($H_1$). We fix the value of learning rate as 0.001 for the experiments.

| Dataset | $\rho = 0.02$ | $\rho = 0.01$ | $\rho = 0.005$ |
|---|---|---|---|
| MUTAG | $87.9 \pm 8.1$ | $87.8 \pm 8.8$ | $87.9 \pm 8.1$ |

Table 4: 10-fold cross-validated test accuracy for different grid resolutions.

In Table 5, we provide a study of the performance of GRIL on different values of $k$ on MUTAG and COX2 datasets. We compare the 10-fold cross-validated test accuracy of GRIL on Density-Alpha bi-filtration. For this study, we use a 1-layer MLP classifier and we fix the learning rate to be 0.001. The columns in Table 5 represent the values of $k$ chosen. For instance, $[1 - 2]$ represents that we computed $\lambda(\boldsymbol{p}, 1, \ell)$ and $\lambda(\boldsymbol{p}, 2, \ell)$ and concatenated these vectors before passing them to the 1-layer MLP classifier. It seems that for datasets with smaller graphs such as MUTAG, using ranks higher than 6 are not very useful. However, for datasets with comparatively bigger graphs, using higher ranks seems to increase the performance of the model.

| Dataset | [1-2] | [1-4] | [1-6] | [1-8] | [1-10] |
|---|---|---|---|---|---|
| MUTAG | $83.1 \pm 9.0$ | $86.8 \pm 7.5$ | $88.4 \pm 8.4$ | $87.8 \pm 8.8$ | $87.8 \pm 8.8$ |
| COX2 | $78.7 \pm 0.0$ | $78.3 \pm 1.2$ | $79.6 \pm 3.0$ | $80.4 \pm 2.2$ | $80.6 \pm 2.5$ |

Table 5: 10-fold cross-validated test accuracy of Gril D-Alpha for different values of $k$.

We report the computation time for computing $\lambda(\boldsymbol{p}, k, \ell)$ where $l = 2$, $k \in \{1, 2, \ldots, 10\}$ and $\boldsymbol{p} \in \mathcal{P}$ where $|\mathcal{P}| = 128$ in Table 6. The GRIL features were calculated on Density-Alpha bi-filtration. The computations were done on a Intel(R) Xeon(R) Gold 6248R CPU machine and the computation was carried out on 32 cores. We report the total computation time per dataset average time it takes for computation time (in seconds) per graph for each dataset.

| Dataset | Total Computation Time (s) | Computation Time per graph(s) |
|---|---|---|
| PROTEINS | 27851 | 25.02 |
| COX2 | 15878 | 34.00 |
| DHFR | 23121 | 30.58 |
| MUTAG | 4358 | 23.18 |

Table 6: Computation times for GRIL features on Density-Alpha bi-filtration for graph datasets

In Table 4, we report the accuracy of GRIL on Density-Alpha bi-filtration with different grid resolutions on the MUTAG dataset.

