# OpenReview forum: "A $2$-parameter Persistence Layer for Learning"
_ICLR.cc/2023/Conference — Submitted to ICLR 2023_

### Official Review · Reviewer_SJck · 2022-10-21

**Confidence:** 5
**Correctness:** 2
**Technical Novelty And Significance:** 3
**Empirical Novelty And Significance:** 3
**Recommendation:** 3

**Clarity, Quality, Novelty And Reproducibility:**

## Clarity

- The abstract discusses the calculation of a cycle basis; I think this could be rephrased to be more accessible to a general audience of machine learning readers.

- When mentioning that 2-parameter summaries need to be generated for machine learning and that this is a relevant problem, consider discussing potential solutions. What about the sliced barcode or the work by [Lesnick and Wright on 2-parameter persistent homology](https://arxiv.org/abs/1512.00180).

- The implications of the discretisation strategy in Figure 2 need to be discussed more. I had trouble following this section upon my first read. In a revision, the implications of parameter choices should be pointed out directly (or references to specific ablation studies should be given).

- Moreover, it is crucial to put *all* definitions that are required to understand the method in the main text. I found it very hard to switch between the main text and the appendix upon trying to establish a mental model of the proposed method.

- The new method requires understanding a slew of concepts, to wit: persistent homology / topological data analysis, 1-parameter persistence modules, 2-parameter persistence modules, persistence landscapes, and, later on, even zigzag persistence. It is therefore relevant to make sure that the "gist" of all these concepts is at least explained briefly. Ideally, all the concepts could be briefly depicted in an overview figure, containing a "working example". I think the paper would immensely benefit from this!

- On a more abstract level, it should be clarified what the benefits of the discretisation are. Especially for readers that have a passing familiarity with the 1-parameter case, which does not typically necessitate

- Figure 4 is missing details and cannot be interpreted easily. I would at least clarify that the bi-filtration is using "density" and "distance", respectively.

- The stability properties (Proposition 2.1 and subsequent statements) could be mentioned before; I think they constitute a great result!

- Why is Remark 2.1 necessary? It seems that this could be handled with Remark 2.2

- Why is the remark on worm construction that the landscape function only checks values on $\mathcal{P}$ necessary? I thought that this was *by definition*.

- Since the sensitivity to resolution is remarked upon, it should be assessed more thoroughly in an ablation study in the main text.

- Algorithm 1 is hard to understand without being aware of zigzag persistence. I would suggest to relegate it to the appendix.

- What is $\omega$ in the complexity discussion? Does it refer to matrix multiplication complexity?

## Quality

- Adding a more detailed delineation to existing research would help assess the quality of the paper even better. For instance, the aforementioned paper by Lesnick and Wright is missing from the discussion altogether even though it constitutes a major step towards better understanding the 2-parameter case (and it also provides computable invariants).

- Adding standard deviations to the experiments is vital. I trust that "GRIL" performs as well as claimed, but it would be interesting to understand to what extent the results are stable with respect to different initialisations.

- In the experimental section, I am wondering to what extent it would be useful to compare to multi-parameter persistence landscapes or sliced barcodes. The current comparison partners are all incapable of leveraging multiple filtrations, so the comparison in its current form seems slightly unfair.

## Novelty

The method as such is novel but some related work on multiparameter persistence modules could be discussed in more detail. I think that work by Lesnick and Wright should be particularly discussed in more detail (see above).

## Reproducibility

The work should be reproducible by an expert in topological data analysis. Additional code would have been appreciate to at least get a quicker understanding of how the method is supposed to work.

## Minor issues

- When using `natib`, please use `\citet` and `\citep` consistently. The former is to be preferred when it comes to in-text citations.

- "summarized as persistence diagram" --> "summarized [in|as] a persistence diagram". Similar issues can be found throughout the text; I would recommend another pass over the text to check for missing articles, etc.

- "specially, for $l = 1$" --> "As a special case, [...]"

- Please check the bibliography for consistency; there are some issues with redundant URLs, ISSNs, capitalisation ("morse theory" instead of "Morse theory"), and old venues for papers (i.e. an arXiv version is cited instead of the published version). An additional pass over the bibliography would be warranted.


**Strength And Weaknesses:**

The main strength of the paper lies in its **novel, original vectorisation strategy**. Bi-filtrations being a natural way of describing many data sets, the new method addresses a clear gap in the literature, and it has the potential to become a strong contribution to the topological data analysis literature. That being said, the paper in its current form suffers from two major weaknesses:

1. *Lack of clarity*: the paper is currently not accessible for a non-expert reader. Given the high complexity of the topic, a more intuitive description of the required methods, as well as some more "hand-holding" of readers, would be necessary. In addition, the write-up is at times quite cursory, and will require a substantial revision.

2. *Lack of experiments*: the experiments shown in the paper do not provide a sufficient depth to appreciate the contributions; the graph classification example, while interesting, would require more explanations concerning the task at hand, while the loss term example would require a comparison to other techniques.

I will subsequently comment more on these two aspects.


**Summary Of The Paper:**

This paper presents a new neural network layer for leveraging 2-parameter persistent homology, i.e. a topology-driven method that is capable of incorporating information from two independent filtration functions (serving as different "perspectives" under which to view data set, such as "density" versus "distance") of a point cloud data set. This is achieved by employing a novel vectorisation of topological features, based on the rank of 2-parameter persistence modules, a way to represent topological features algebraically. The proposed method can be used in a supervised setting, which is demonstrated by means of classifying graphs, as well as an unsupervised setting (serving as a loss term for adjusting the shape of point clouds according to their topological features.


**Summary Of The Review:**

While I appreciate the novel and creative direction taken by the paper, I cannot endorse it for publication in its present form. The changes that are required to improve it cannot be done within a conference cycle and necessitate a major revision. I understand that this is not the desired outcome for the authors, but I want to stress that I believe that this paper has the potential make a substantial contribution to the field, provided the two issues mentioned above are addressed.

---

### Official Review · Reviewer_HJRc · 2022-10-24

**Confidence:** 4
**Correctness:** 3
**Technical Novelty And Significance:** 2
**Empirical Novelty And Significance:** 2
**Recommendation:** 5

**Clarity, Quality, Novelty And Reproducibility:**

This article is overall well written. The proposed approach is based on an already existing object (the generalized rank invariant), but is novel in how it handles and compute it for practical computations.

**Strength And Weaknesses:**

Strengths:
---This vectorization uses more information from the module than most of the current ones, which only use the fibered barcodes.
---This pipeline is stable and differentiable with respect to the input.
---This generalizes the multiparameter persistence landscapes (MPL), when l=1.
---This construction relies on the generalized rank invariant, which can be efficiently computed for 2-persistence modules. Furthermore, each component of this vectorization can be computed independently, which makes the computation easily parallelizable. In particular, it may be scalable to larger datasets (but it would require more experiments).

Weaknesses:
The major weaknesses come from the numerical experiments.
---The authors only provide synthetic data sets, with small numbers of simplices, which makes it difficult to assess how scalable and efficient the approach is in practice
---As far as I understand, when l=1, the proposed approach is equal to the MPL, however there is no score comparison (running time / accuracy tradeoff wrt to the parameter l)
---Although the classification performances seem to be positive, the authors only compare to the performance of the persistence image, which 1) is not always the best persistence diagram vectorization (it would be good to also try landscapes, silhouette and kernels such as the sliced Wasserstein kernel), and 2) is not a 2-parameter persistence vectorization (it would be nice to compare also to the multiparameter persistence image of Carriere and Blumberg and the multiparameter persistence landscape of Vipond).
---Why are the accuracy provided with no variances?

**Summary Of The Paper:**

The paper proposes an approach of vectorization of 2-parameter persistence modules, based on the generalized rank invariant, computed over so-called worm-shaped 2-intervals. The authors show that this construction is stable w.r.t. the interleaving distance, and that this construction can be differentiated w.r.t. the bi-filtration used to construct the 2-persistence module, when the latter is 1-critical. Finally they provide some experiments on synthetic data sets.

**Summary Of The Review:**

Overall I think that this work is encouraging, but the lack of practical experiments makes the performance improvements uncertain.

---

### Official Review · Reviewer_Jgx9 · 2022-10-27

**Confidence:** 4
**Clarity, Quality, Novelty And Reproducibility:** The presentation is reasonably clear.…
**Correctness:** 2
**Technical Novelty And Significance:** 3
**Empirical Novelty And Significance:** 2
**Recommendation:** 6

**Strength And Weaknesses:**

Generally, I like the paper. It is well written considering how challenging the topic is. Multiparameter persistence is very important and can be quite useful in practice. The idea of landscape is well aligned with data analytics purpose. I tend to believe the idea will work in practice. However, I am holding my scores below the bar due to several concerns regarding the experiments.

The experiment section, in my opinion, should be showing that the representation is useful and delivers rich information than existing representations. But this is not thoroughly conveyed in the experiment section. The experiment has two parts: landscape as a static feature and as a differentiable layer.

As a static feature,
1, I am not sure why only the HourGlass data is used when plenty of previous papers (Hofer et al, PersLay (Carrier et al), etc) used public graph classification benchmarks. The dataset is interesting. But to show the representation is useful, previously used graph classification benchmarks should be used.

2, I am not really worried about comparing to SOTA GNN methods. But the proposed representation should outperform existing persistence-based representations. This includes not only persistence image, but also 1-parameter persistence kernels and some recent 2-parameter persistence kernels/representations. As long as these comparisons can be shown on datasets like REDDIT, IMDB, etc, used in Hofer et al. and PersLay, I would be convinced that the proposed landscape is indeed a better representation.

The differentiable layer experiment is an interesting proof-of-concept with regard to a specific manually chosen loss function. In this sense, emphasizing it as a main benefit (and thus the title) is a bit of an oversell. To me, the contribution is sufficiently important as long as the representation can be proven as a rich feature in the static experiments (with benchmarks and stronger baselines).

Minor comments:
1) end of the first paragraph - "From the perspective of direct use of 2-parameter persistence modules into ML models, to the best of our knowledge, is the first of its kind". I am not sure i understand this, isn't previous 2-parameter persistence kernels already doing this? Or you meant you are the first to use the non-slicing version of 2-parameter persistence in ML? Please elaborate.

2) it would be helpful to make it more explicit the novel contribution of this paper compared to Kim and Memoli.

3) Definition 2.1 could have been better illustrated with more examples considering this is one of the key contributions. The examples in Fig 2 and its caption is quite short.

4) in complexity analysis, shouldn't $t$ be $n^2$ in the worst case? What is $\omega$? Is it the exponent of the matrix multiplication complexity? If so, please also discuss the complexity of a practical implementation of the method (if I understand correctly, matrix multiplication complexity for persistence is only theoretical).


**Summary Of The Paper:**

This paper proposes a vector representation of a 2-parameter persistent homology. This is a very important problem. Application of multiparameter persistence can have significance impact in developing topology-based learning methods.

The proposed method is based on the generalized rank invariance by Kim and Memoli and is inspired by the classic persistence landscape on 1-parameter persistence by Bubenik. Theoretical results (stability and differentiability, etc) are provided, although they are not particularly surprising given the known nice properties of rank invariance and 1-parameter landscapes. An algorithm is proposed to compute the proposed representation. Experimental results on a synthetic dataset is used to show that the representation can be used as topological feature for undirected graphs, and can be used as a differentiable layer for end-to-end learning.



**Summary Of The Review:**

Overall, I think the problem is important and the proposed idea is sufficiently significant. I am holding my scores at this moment because the experimental evaluation is not sufficient to support the main claim.

---

### Official Review · Reviewer_nKKg · 2022-10-27

**Confidence:** 3
**Correctness:** 2
**Technical Novelty And Significance:** 3
**Empirical Novelty And Significance:** 2
**Recommendation:** 3

**Clarity, Quality, Novelty And Reproducibility:**

## Clarity

The paper suffers from some drawbacks in terms of clarity. The introduction is well-written in my opinion, but things get confusing afterwards.

## Quality

The paper addresses an arguably very difficult problem and attempts to give a computational solution to it. However, it suffers too many drawbacks that hinder its quality in my opinion.

## Novelty

The approach is novel to the best of my knowledge.

## Reproducibility

The paper suffers on the reproducibility side (which, of course, do not mean that the work is not correct). Theoretical statement are not very well explained/introduced, rely on many concepts that are only quickly discussed in the appendix, and I think that the main body is insufficient for proofreading by someone who is not already very familiar with 2-persistence (and zigzags).

From the experimental side, some crucial details are only deferred to the appendix, and some are missing (e.g. running times, discussion on influence of parameters, etc.), which only give a limited understanding of the potential impact of PersGril in practice.

**Strength And Weaknesses:**

## Strengths

- Multi-parameter persistence is an extremely challenging topic (from both a computational and theoretical side), and there are very few practical tools developed to handle it (and most of them essentially rely on computing 1d-persistence). To that respect, any improvement in that direction---as the one proposed in this work---is worth of interest.

## Weaknesses

- The experimental section is pretty limited: the practical performance of PersGril is only showcased on a toy dataset, and while the results are reasonably convincing at showing that PersGril is capable of doing great on a dedicated dataset, I would not say that these experiments truly showcase the use of 2-parameter persistence in machine learning tasks. It is quite below, for instance, the experiments conducted in the work of Carrière and Blumberg, NeurIPS 2020, that also leverage 2-parameter persistence---and the current work does not compare to it (if the comparison is irrelevant, please discuss it). In addition,
   - The presentation of the experiments lacks details. For instance, PersGril outputs some vector (one real number of each $p,k,\ell$). How is the actual classification performed afterward? [note: ok I found the answer in Section C.0.1. It's a 3-layer perceptron. This should be explained in the main body directly.]
   - There is no explicit running time reported. While Time Complexity is discussed, practical running time is also important (if not more) to give an idea of how practical the proposed method is. If PersGril is computationally expensive to run, this should be discussed.
   - The influence of parameters (noting that there values are only discussed in Section C.0.1) is not discussed. For instance, we learn in C.0.1 that $p$ ranges on a $4 \times 4$ grid. What's the influence of taking a $3 \times 3$ grid, and a $5 \times 5$ one in terms of both computational time and model accuracy? (Note : since the model achieves perfect accuracy on this dataset, this may not be very enlightening. Showcasing it on a harder dataset would be more interesting).

- Despite some effort that have been made in the presentation (e.g. in the introduction), the paper is quite hard to understand in details. In particular the description of the algorithm, in Section 3, is hard to parse, especially for the reader that is not familiar with zigzag persistence (which, arguably, is likely to be a general case). I would think that a step-by-step example (at least in the appendix, if space constraints do not allow for it) would allow the reader to understand "what is going on". Similarly,
   - The "density-Rips filtration" in Figure 4 has not been defined as far as I can tell. Rips filtration should be quickly defined at least in the appendix (saying it's the 1d filtration induced by the map $f : x \mapsto \mathrm{dist}(x, P)$ where $P$ denotes the point cloud should be enough). Also, being unfamiliar with 2-parameter persistence, I struggle to understand the second plot. Why do points organize along discrete vertical lines? It seems that some of the parameter (density I guess?) has only been taken at discrete steps (0.1), is that correct ?
   - The distance $d_I$ used in the proof of Proposition 2.1 in the appendix has not been defined as far as I can tell. I guess it is some sort of interleaving distance, but I'm not sure.
   - What is the exponent $\omega$ in the Time Complexity paragraph? (I may have miss its definition, but a quick ctrl-F suggest that this symbol is only used there.)
   - I don't think that section C.0.1 should belong to the appendix. At least some of it should belong to the description of the dataset. The only thing that the main body says about the two classes is that "they essentially have the same structure", which is not very instructive. What are we trying to classify here? Reading the appendix should not be necessary to understand such central points of the work.

**Summary Of The Paper:**

This article introduces a vectorization---called PersGril---of $2$-parameter-based persistence, akin to the well-known persistence landscapes routinely used for 1-persistent homology. They prove that their vectorization is stable (1-Lipschitz for the infinite norm) with respect to the inputs functions $\mathcal{X} \to \mathbb{R}^2$, hence its differentiability (a.e.).

They derive a practical algorithm that rely on zigzag persistence (a sort of intermediate between 1 and 2 persistence, loosely) for which existing software already exists.

The showcase their approach on few numerical experiments

**Summary Of The Review:**

While I would like to see the development of more (computational) tools dedicated to multi-persistence---and in that regard I am quite positive about the proposed approach---I think that this work has some caveats in terms of (i) clarity of the presentation, (ii) experimental descriptions, that prevent me to support its publication for now.

---

### Decision · Program_Chairs · 2023-01-20

**Decision:**

Reject

**Justification For Why Not Higher Score:**

The paper has received interest from reviewers, but too many areas are left for improvements to ensure this is published as it is.

**Justification For Why Not Lower Score:**

NA

**Metareview: Summary, Strengths And Weaknesses:**

The authors propose a new TDA tool that uses 2 parameters (rather than just one) to compute feature representations of topological data. All reviewers mention that the idea is interesting, but suffers from a few important flaws. Writing is one (too condensed, and not necessarily easy to parse for the ICLR audience) but most of the criticism was geared towards the experiments, notably in the original version. Some were added during the rebuttal but reviewers still feel an additional round is needed. I believe the "proof of concept" experiment on differentiability is a two edged sword. While it might make the work more appealing, it also raises the question of why, given this is ICLR and not a TDA focused venue, this is not more clearly emphasized in the experiments. I also think "differentiability" can have various meanings, here the authors use the minimal one (a.e.) which raises the question of how this would perform in practice. I think all these aspects require a new reviewing round.